# Oral Microbiota as Novel Biomarkers for Colorectal Cancer Screening

**DOI:** 10.3390/cancers15010192

**Published:** 2022-12-28

**Authors:** Sama Rezasoltani, Hamid Asadzadeh Aghdaei, Seyedesomaye Jasemi, Maria Gazouli, Nikolas Dovrolis, Amir Sadeghi, Hartmut Schlüter, Mohammad Reza Zali, Leonardo Antonio Sechi, Mohammad Mehdi Feizabadi

**Affiliations:** 1Basic and Molecular Epidemiology of Gastrointestinal Disorders Research Center, Research Institute for Gastroenterology and Liver Diseases, Shahid Beheshti University of Medical Sciences, Tehran 19835-178, Iran; 2Section Mass Spectrometry and Proteomics, Institute of Clinical Chemistry and Laboratory Medicine, University Medical Center Hamburg-Eppendorf (UKE), 20246 Hamburg, Germany; 3Microbiology Section, Department of Biomedical Sciences, University of Sassari, Viale San Pietro 43b, 07100 Sassari, Italy; 4Department of Basic Medical Sciences, Medical School, National and Kapodistrian University of Athens, 11527 Athens, Greece; 5Laboratory of Biology, Department of Medicine, Democritus University of Thrace, 68100 Alexandroupolis, Greece; 6Gastroenterology and Liver Diseases Research Center, Research Institute for Gastroenterology and Liver Diseases, Shahid Beheshti University of Medical Sciences, Tehran 19835-178, Iran; 7Department of Microbiology, School of Medicine, Tehran University of Medical Sciences, Tehran 19835-178, Iran

**Keywords:** colorectal cancer, oral microbiota, biomarkers, non-invasive diagnosis, metagenomics

## Abstract

**Simple Summary:**

Colorectal cancer (CRC) is the third most common cancer type worldwide. Increasingly, the gastrointestinal microbiome has been identified as an important factor in the pathogenesis of CRC. In this work, for the first time in Iran, an analysis of the whole microbiome in saliva and stool samples of CRC patients has been conducted, and the results were investigated and compared with healthy controls (HCs). In the present study, we found that there is a distinct clustering of genera associated with the saliva and fecal samples of CRC patients and HCs, respectively, pointing to special microbial signatures in both conditions. However, the roles of microbiota in CRC and HCs require further investigation.

**Abstract:**

Alterations of the gut microbiome in cases of colorectal cancer (CRC) hint at the involvement of host–microbe interactions in the onset and progression of CRC and also, possibly, provide novel ways to detect and prevent CRC early. The aim of the present study was to evaluate whether the oral and fecal microbiomes of an individual can be suitable for CRC screening. Oral and fecal samples (*n* = 80) were gathered in Taleghani hospital, affiliated with Shahid Beheshti University of Medical Sciences, Tehran–Iran, from CRC stage 0 and I patients and healthy controls (HCs), who were screened for the first time. Microbial metagenomics assays were performed for studying microbiota profiles in all oral and fecal samples gathered. An abundance of top bacterial genera from both types of specimens (fecal and saliva samples) revealed a distinction between CRC patients and HCs. In saliva samples, the α diversity index was different between the microbiome of HCs and CRC patients, while β diversity showed a densely clustered microbiome in the HCs but a more dispersed pattern in CRC cases. The α and β diversity of fecal microbiota between HCs and CRC patients showed no statistically significant differences. *Bifidobacterium* was identified as a potential bacterial biomarker in CRC saliva samples, while *Fusobacterium*, *Dialister*, *Catonella*, *Tennerella*, *Eubacterium-brachy-group*, and *Fretibacterium* were ideal to distinguish HCs from CRC patients. One of the reasons for the heterogeneity of CRC may be the gastrointestinal (GI) tract microbiota, which can also cause systematic resistance to CRC. Moreover, an evaluation of saliva microbiota might offer a suitable screening test for the early detection of this malignancy, providing more accurate results than its fecal counterpart.

## 1. Introduction

Colorectal cancer (CRC) is ranked third in terms of recognition and second in terms of mortality among the most important malignancies in the world, with an increasing trend in incidences each year [1,2]. Many Asian countries including Iran have seen an increased rate in the incidence of CRC during the last decade [3]. When CRC is detected in the early stages, the five-year survival rate is above 80%, while, in advanced stages, this rate reaches less than 10% [4]. Today, the fecal immunochemical test (FIT), as a non-invasive screening method, is applied to find high-risk individuals who are then referred for CRC screening. However, the sensitivity of this test is low and produces variable results among different studies and populations [5]. Hence, more accurate and non-invasive screening is needed to determine patients with early stages of CRC. Over the years, the relationship between gut microbiota and various diseases has become the focus for many researchers [6,7], including those studying CRC [8].

In CRC cases, altered gut microbiome hints at a possible role of the host–microbial interaction in the initiation and progression of CRC [4,9]. The changes of gut microbial composition may be considered as an accurate test in CRC screening with appropriate specificity and sensitivity [10,11]. Previously, our team, among others, have shown changes of colonic mucosal and fecal microbiome in CRC cases [12,13,14,15,16,17] and also the suitability of fecal microbiome analysis as a powerful method for CRC diagnosis [9,10,14], especially in combination with the FIT [10]. In addition, the distinct profiles of microbiota in saliva have been previously correlated to oral [18,19], esophageal [20], and pancreatic cancers (PCs) [21,22]. In addition, the microbiome related to the oral cavity can also be found in the mucosa and feces and is associated with CRC [4,11,12,13,14,23,24]. This encouraged us to evaluate oral microbiome in CRC as a viable alternative and to determine whether saliva sampling can be used as a convenient and readily available methodology for CRC screening.

In this work, our aim was to utilize metagenomics analyses in order to compare the oral and fecal microbiome composition and diversity between CRC patients and healthy controls (HCs) in the Iranian population. This has allowed us to develop a classifier of saliva and fecal microbiome assessment with the goal of finding novel biomarkers for noninvasive CRC screening. This study also supplements the current literature in which there is a lack of studies regarding gut microbiota in several underdeveloped countries.

## 2. Materials and Methods

### 2.1. Human Oral Sampling

Saliva samples (*n* = 40) were gathered, in the current case-control investigation from Taleghani hospital, affiliated with Shahid Beheshti University of Medical Sciences, Tehran–Iran, consisting of CRC TNM stage 0, I and HC groups (between 2020 and 2021). Saliva specimens were collected between 08:00 am and 12:00 noon, and eating and drinking at least 2 h prior to sampling was not permitted. Saliva was provided in a 20 mL falcon tube on ice. About 1 mL of unstimulated saliva was collected over 5–10 min. Then, specimens were transferred to new 2 mL microtubes and frozen at −80 °C for further evaluation. Cases were precisely identified using the colonoscopy procedure and histopathology results of biopsies.

### 2.2. Human Fecal Sampling

Fecal samples were taken from the same patients and healthy individuals whose saliva was collected; (*n* = 40) including CRC stage 0 and I and HC groups. As above, cases were precisely identified by colonoscopy procedure and histopathology results of biopsies.

### 2.3. Inclusion and Exclusion Criteria

As mentioned in previous studies, the inclusion criteria were individuals presenting changes of bowel habit, rectal bleeding, abdominal pain or anemia, and asymptomatic individuals aged 50 or above undergoing screening colonoscopy [11]. The exclusion criteria for HCs and CRC patients were the use of antibiotics within the past 3 months; a vegetarian diet; an invasive medical intervention within the past 3 months; a past history of any cancer, or inflammatory intestinal disease [11,14].

### 2.4. DNA Extraction and Purification from Oral and Fecal Specimens

Oral specimens were thawed, and Genomic DNA was extracted using the QIAamp DNA Microbiome Kit from Qiagen (Hilden, Germany) according to the manufacturer’s instructions. Fecal samples were thawed on ice and DNA extraction was performed using the QIAamp DNA Fecal Mini Kit according to manufacturer’s instructions (Qiagen). Extracts were then treated with DNase-free RNase to eliminate RNA contamination. DNA quality and quantity were determined using a NanoDrop2000 spectrophotometer (Thermo Fisher Scientific, Waltham, MA, USA).

### 2.5. PCR Amplification and Sequencing

The gene-specific sequences applied in the current study target the 16S ribosomal RNA V3 and V4 regions using two primers: a forward (5′TCGTCGGCAGCGTCAGATGTGTATAAGAGACAGCCTACGGGNGGCWGCAG 3′) and a reverse (5′GTCTCGTGGGCTCGGAGATGTGTATAAGAGACAGGACTACHVGGGTATCTAATCC3′). The PCR (25 µL) was set up as follows: 12.5 µL per sample 2xKAPA HiFi HotStart Ready Mix, 5 µL primer of forward (1 µM), 5 µL primer of reverse (1 µM), and 2.5 µL bacterial genomic DNA (5 ng/µL in 10 mM Tris pH 8.5) in a 96-well 0.2 mL PCR plate. The thermal cycling situation for amplification of PCR was as follows: initial incubation step at 98 °C for 3 min, then 30 denaturation cycles at 94 °C for 30 s, annealing at 55 °C for 30 s, extension at 72 °C for 30 s, and a final extension step at 72 °C for 5 min. Next, 1 µL of PCR product was run on a BioanalyzerDNA 1000 chip to confirm the size. Utilizing the V3 and V4 primer pairs in the study, the expected size on a Bioanalyzer trace after the Amplicon PCR step is ~550 bp.

Amplicon product purification was performed with AMPure XP beads based on the manufacturer’s recommendations to remove all remaining contaminants or PCR artifacts. Purified amplicons were applied to construct the library based on standard protocols, and sequencing was performed using the Nextera XT Index Kiton on an Illumina NovaSeq platform (Illumina, San Diego, CA, USA).

### 2.6. 16S rRNA Sequence Preprocessing and Analysis

Demultiplexed raw sequences were imported into QIIME2 v.2022-2 [25] and were denoised and clustered using DADA2 [26]. Taxonomy classification was conducted using the pre-trained, via scikit-learn [27], SILVA [28] with 138 99% full-length sequences. The resulting amplicon sequence variant (ASV) table, taxonomy assignment, and appropriate metadata were used as input for the Marker Data Profiling module of the online platform Microbiome Analyst [29]. Features with low counts (<4 and <20% prevalence in samples, *n* = 1815) along with those with low variance (based on interquartile range, *n* = 25) were excluded from the downstream analyses counts were normalized using Total Sum Scaling (TSS); moreover, based on their poor quality, 2 samples (1 saliva and 1 fecal, both from the same patient) were excluded from the dataset. Alpha diversity (α-diversity) metrics were calculated using the Chao1 index, and the statistical differences between groups were explored using the Mann–Whitney test. Beta microbial diversity (β-diversity) was analyzed using the non-metric multidimensional scaling (NMDS) ordination method, the Bray–Curtis distance, and Analysis of Similarities (ANOSIM). Both alpha and beta microbial diversity analyses were applied on the feature level. Clustering heatmaps were produced on the genus level using the Ward clustering and Euclidean distance methods. Univariate analysis for detecting differentially abundant features between conditions was performed on the genus level (on raw and log-transformed counts) using the Mann–Whitney test, filtering for results with adjusted *p* values less than 0.05 (p.adjust < 0.05). The Linear Discriminant Analysis (LDA) Effect Size (LEfSe) method [30] was employed to detect genera with biomarker potential. Finally, Random Forest (RF) classification was applied using 1000 tree permutations to test for genera that can serve as explanatory variables for the models. The models’ robustness was evaluated using a calculation of the Out-Of-Bag (OOB) error [31] rates.

## 3. Results

The oral and fecal microbiota composition from CRC patients and HCs was identified and quantified as previously described in order to be used as an input for further analyses.

### 3.1. Microbial Community Composition

Differences between saliva and fecal specimens were investigated by focusing on the composition of the bacterial population and the relative abundance of each genus.

The relative abundance at the genus bacterial level (%) of both types of specimens (fecal and saliva samples) revealed similar but perturbed profiles between CRC patients and HCs (Figure 1). In detail, in the fecal specimens of CRC cases, *Bacteroides* had the highest abundance across genera with 28.61%, followed by *Escherichia_Shigella* (18.68%), *Faecalibacterium* (7.51%), and *UGG-002* (4.83%), while, in the HCs’ individual fecal samples, *Bacteroides* was the most abundant feature, with an average relative abundance of 31.49%, followed by *Escherichia-Shigella* (16.28%), *Dialister* (10.32%), *Roseburia* (5.77%), and *Faecalibacterium* (4.79%).

Moreover, in CRC patient saliva samples, *Streptococcus* (34.34%) was the most abundant genus, followed by *Prevotella* (17.31%), *Veillonella* (14.43%), *Neisseria* (11.61%), and *Rothia* (5.91%). Finally, in the HCs’ individual saliva samples, *Streptococcus* (42.10%) *Veillonella* (15.28%), *Prevotella* (14.55%), *Neisseria* (10.60%), and *Haemophilus* (4.06%) were the most frequent taxa.

### 3.2. Microbial Community Diversity

α-diversity index differentiated between the microbiome of HC saliva compared to CRC patient saliva and was found to be statistically significant using the CHAO1 method (*p* = 0.003) (Figure 2A) but not between the CRC and HC fecal samples (*p* = 0.56) (Figure 2B). A direct comparison of α-diversity between the fecal and saliva samples of CRC patients only also yielded statistical significance (*p* = 0.03) (Figure 2C). As a baseline, the α-diversity of fecal and saliva samples of HCs was calculated and revealed that the saliva samples of HCs are statistically and significantly (*p* = 0.0002) different than HC fecal samples (Figure 2D). Comparing biodiversity in raw numbers between the tested groups resulted in a several observations. HC saliva samples are more biodiverse than CRC saliva ones, whereas CRC fecal samples appear to be more enriched than their HC counterparts. In addition, fecal samples appear to be richer in microbial taxa than saliva samples in CRC patients, whereas, in HCs, these findings are reversed.

NMDS analysis using the Bray–Curtis distance was applied to study saliva microbial β-diversity in HCs compared to CRC patients, and it depicted a dense clustering of the microbiome in HC saliva but more dispersed patterns in CRC saliva, while significant overlaps were detected between the microbiome diversity clusters in CRC with the HC of both sample types (Figure 2E,F). On the other hand, the β-diversity differences between saliva and fecal in both CRC and HC samples are distinct and reveal unique localized microbiomes (Figure 2G,H).

### 3.3. Phenotype–Microbial Associations

Clustered heatmaps, for differentially abundant bacterial genera in the saliva of CRC patients and HCs (Figure 3A) and the fecal samples of CRC patients and HCs (Figure 3B), were created. There appears to be no clear clustering of genera associated with CRC or HC in the saliva (Figure 3A) or fecal samples (Figure 3B), respectively; however, certain strong associations can be ascertained in the smaller subclusters of samples.

On the other hand, in the clustered heatmaps of genera–group associations, and using all samples, regardless of phenotype and collection source, there appears to be a clear clustering of genera associated with saliva and fecal samples, respectively, whereas the condition appears to affect composition only (Figure 4). These findings appear to be in agreement with the previous β-diversity observations.

### 3.4. Classification and Biomarker Discovery

By employing a Random Forest (RF) model, a robust supervised classification algorithm was created to detect features which can differentiate between phenotypes and provide some predictive power. The analyses were based on bacterial genera relative abundances in the saliva of CRC patients and the HC group (Figure 5A) and in the fecal samples of CRC individuals and the HC group (Figure 5B). This has allowed for testing of the respective predictive powers of saliva and fecal microbiomes in CRC detection while highlighting specific features (microbial genera) that are important for each model. The model based on the saliva microbiome outperformed the fecal one based on the OOB error of the models (21.6% for saliva and 43.6% for fecal) along the correctly classified samples. The saliva model highlighted features such as the *Eubacterium_brachy_group*, *Bifidobacterium*, *Fusobacterium*, *Catonella*, and others while the fecal model based its classification decisions on features such as the *Ruminococcus_torques_group*, *Collinsela*, *Ruminococcus_gauvreauii_group*, *Monoglobus*, and *Parabacteroides*.

To supplement and further validate the classification approach, Linear discriminant analysis Effect Size (LEfSe) was used to detect and highlight potential diagnostic microbial biomarkers. From the findings of the LEfSe for the saliva microbiome in CRC and the HC (Figure 5C), a total of seven genera were highlighted: six in HC saliva and one in CRC saliva, which can help us differentiate between phenotypes. The HC saliva samples were identified using the genera *Fusobacterium*, *Dialister*, *Catonella*, *Tennerella*, *Eubacterium-brachy-group* and *Fretibacterium*, while the CRC saliva was characterized by the genus *Bifidobacterium*.

A similar analysis on the fecal microbiome in CRC and the HC (Figure 5D) showed a total of six genera, three of which are associated with HC fecal and three with CRC fecal. As depicted in Figure 5D, HC fecal samples were identified using the genera *Parabacteroides*, *Barnesiella*, and *Collinsella*, while the CRC fecal samples were characterized by the genera *Ruminococcus-torques-group*, *Granulicatella*, and *Ruminococcus-gauvreauii*-group.

## 4. Discussion

The present study describes a combined analysis of the microbiome in saliva and fecal samples of CRC Iranian patients. Using abundance metrics, classification models, and biomarker discovery approaches, based on bacterial genera relative abundances, in the CRC saliva, HC saliva, CRC fecal, and HC fecal samples, our findings indicate certain features as potential bacterial biomarkers for the early diagnosis of CRC.

In the present study, we found that there is a clear clustering of genera associated with the saliva and fecal samples of CRC patients and the HC, respectively, pointing to distinct microbial signatures in both conditions. The most abundant microbial genera among saliva specimens were *Streptococcus* followed by *Prevotella*, *Veillonella*, *Neisseria*, and *Rothia* in CRC cases, while, in the HC, *Streptococcus* followed by *Veillonella*, *Prevotella*, *Neisseria*, and *Haemophilus* were the most frequent. Our analysis of similarity between saliva and fecal samples in CRC and the HC showed distinct bacterial taxonomic compositions. Even though, statistically, the differences on the genera level do not show any significance between CRC and HC using univariate methodologies, they highlight the distinct compositional profiles of saliva and fecal samples and point to their individual ability to provide unique insights during dysbiosis. These findings also underline the need for the more complex statistical analyses, such as the LEfSe and ML approaches employed in our subsequent analyses. The latter analyses also reveal the top three genera that were differentially abundant–*Eubacterium* spp., *Bifidobacterium* spp., and Fusobacterium spp–in the saliva of CRC patients vs. healthy controls. *Eubacterium* and *Fusobacterium* species might contribute to cancer initiation via promoting inflammation [32]. *Bifidobacterium* species might exhibit anticancer activity on CRC by decreasing and boosting anti-apoptotic and pro-apoptotic genes [33]. These differences in saliva microbial composition between the CRC and controls further strengthens the theory that saliva microbiota could be a possible diagnostic biomarker in the future. Regarding the fecal samples, the top three differentially abundant genera were the *Ruminococcus torques* group and *Collinsella* and *Ruminococcus gauvreauii* group. Our results are in agreement with previous findings [34,35,36] that suggest that *Ruminococcus* 2, the *Ruminococcus torques* group, and *Collinsella* could be novel fecal biomarkers for CRC diagnosis. The low OOB error values exhibited by our predictive models encourages the use of saliva-based prediction for the occurrence of CRC. Previous studies [23,37,38,39] are also in agreement with our results. Furthermore, Boleij et al. reported an increased risk of CRC and colorectal adenomas particularly in patients with *Streptococcus*-associated endocarditis and bacteremia [40]. Additionally, our results supported the findings of Guven et al. [37] that also reported a higher amount of *Streptococcus* in CRC patients, even if the role of *Streptococcus* in the carcinogenesis process is debatable. There are studies supporting the idea that *Streptoccocus* is implicated in the inflammatory process of CRC pathogenesis [41,42]; however, there are those that support the idea that Streptococcus is more likely an opportunist pathogen benefiting from a favorable oncogenic environment [43]. Nevertheless, the increased amount of *Streptococcus* in the saliva of CRC patients is an interesting finding. Furthermore, the increased abundance of *Prevotella*, *Veillonella*, *Neisseria*, and *Rothia* in cancer saliva samples has been confirmed in a number of studies [44,45], indicating their potential contribution in carcinogenesis process.

Regarding the fecal samples compared to the literature, we surprisingly found that the CRC patients were more diverse than the HC in all alpha diversity indices [46]. This observation may be related to ethnic nutritional habits and the genetic background, since it has been observed in the Iranian CRC population. However, it is well known that changes over time in the gut microbiome can occur during the course of the disease and of therapy and in other types of cancers (i.e., metastatic kidney cancer, cervical cancer); moreover, it has been reported that the patients with the highest benefit from cancer treatment were those with higher microbial diversity [47,48]. For example, the increased biodiversity of the cervical microbiome is associated with cervical cancer [48].

Another notable finding of our research is that the saliva microbiome can predict CRC more accurately than the fecal microbiome. Six genera in HC saliva samples and one in the CRC saliva samples can accurately differentiate the two phenotypes. CRC saliva samples were characterized by the genus *Bifidobacterium*, a finding which is supported by previous studies that reported, in mouth rinse samples, oral pathogenic taxa such as *Treponema denticola*, *Bifidobacteriaceae*, and *Prevotella* (*P. denticola*, *P. intermedia*, *P. oral taxon 300*), which were positively associated with an increased risk of CRC [49].

Our study has some limitations that need to be considered when interpreting the results. First, the sample size of this work can potentially be hindering since no statistical power calculations had been applied when designing the study, which could have led to more accurate conclusions. Moreover, based on the current study, α and β diversity of fecal microbiome in the HC and CRC patients showed no statistically significant differences, suggesting that the microbial diversity and abundance in the HC group are the same as in the CRC group. Flemer et al. confirmed these results regarding the microbiota in fecal samples [23]. They claimed that similar bacterial networks in CRC or HC colon biopsies and saliva samples indicate that these bacterial networks existed before CRC progression and can hypothetically be associated with CRC development. Furthermore, they reported that these bacterial communities were just partially distinguishable in the fecal specimens of healthy people or those with CRC, which proves that these networks establish a strong and tight relationship with the intestinal mucosa, which shows the limitations of examining fecal to identify the microbiome in CRC [23]. In addition, the fact that only stool samples were considered, and not those associated with the intestinal mucosa, possibly limits our understanding of the CRC–microbiota link, which has been shown in the Flemer et al. study to differ in diversity and enrichment from fecal samples. Similar bacterial networks from the oral cavity were found more in the colon tissues than in the fecal samples. In fact, colonic mucosal samples help to better detect gut microbiota, although its preparation is invasive, while the fecal samples are easily obtained and its evaluation is considered non-invasive [50]. Our future studies will merge oral, mucosal, and fecal samples to better clarify whether saliva bacterial communities exist before CRC progression and if they can hypothetically be responsible for CRC formation. Finally, while we found alterations of the saliva and fecal microbiota compositions, our approach was based on 16S rRNA amplicons instead of using metagenomic sequencing, thus limiting our capability to identify specific bacteria at the species level and report on bacterial function and their potential metabolites, whose mechanisms of function are uncertain.

It must be noted that this non-invasive diagnostic test, which contains the results of the microbiome of saliva and feces, can increase sensitivity in combination with the non-invasive FIT. It is well known that FIT is able to only identify advanced CRC stages (III and IV) by tracing the blood from intestinal lesions into the feces and that it is not able to identify primary lesions and CRC in stage 0, I and II with sufficient sensitivity [5]. Moreover, the identification of methylated and tumor DNA in fecal samples in addition to occult blood detection seems to be a promising strategy to increase the sensitivity of FIT [51]. However, there are disadvantages in these methods, such as the complicated requirement of fecal collection, the technical complexity requiring multistep lab analysis, and the false-positive rate, leaving colonoscopy as the gold standard tests for screening. Thus, new non-invasive strategies, to the benefit of patients, need to be developed in order to improve CRC screening [52], and the present work provides a possible solution. In our previous works, we investigated the fecal microbiota in cases with various types of colon polyps and reported that the gut microbiome can intervene in CRC early stages through adenoma polyps (AP) development but not hyperplastic polyp (HP) and sessile serrated adenoma (SSA). We also found that AP is an intermediate stage between healthy people and those with CRC, therefore gut microbiota can be considered as possible biomarkers for CRC early detection that may occur later in these patients [14,17]. Based on those results, we concluded that the study of healthy microbiota that are reduced in patients with AP might promote the implementation of nutritional intervention and prebiotic and probiotic treatments to stimulate their growth again some years before the development of this malignancy [14,17]. However, oral cavity microbiota also plays an important role in maintaining homeostasis and may indicate the oral and general health status. The significant differences in the saliva and fecal microbiota of CRC patients, compared to the HC, would provide new insights for the disease pathogenesis and prognosis.

As we mentioned above, the sensitivity of FIT is low and it could miscategorize one-third of CRC 0, I, and II stages [5]. Therefore, more reliable biomarkers are required besides the FIT.

One of the strong points of this study is that the samples were collected in Taleghani hospital, which is the second center of the country for GI disorders patients. Hence, this study is population-based, and all people who had digestive problems and came to this institute for their first visit were considered for our investigation. Therefore, group studies had higher background similarities to each other. As genetics, lifestyle, dietary habits, and body mass index (BMI) differ between various people, the microbial biomarkers identified in the current research must be tested in other studies for validation.

## 5. Conclusions

Our results revealed that saliva microbiome can be considered as a novel biomarker in CRC’s early diagnosis. If the applicability of the saliva microbiome for early CRC diagnosis could be confirmed in a larger population, this might permanently progress the present screening program and subsequently impact clinical practice and guidelines. Additional studies exploring microbial richness and diversity based on lifestyle choices such as eating habits, smoking, and exercise are necessary to enhance the concept of “healthy” living and its impact on CRC prevalence, as well as to promote the finding of novel diagnostics and open up new interventional treatment approaches.

## Figures and Tables

**Figure 1 cancers-15-00192-f001:**
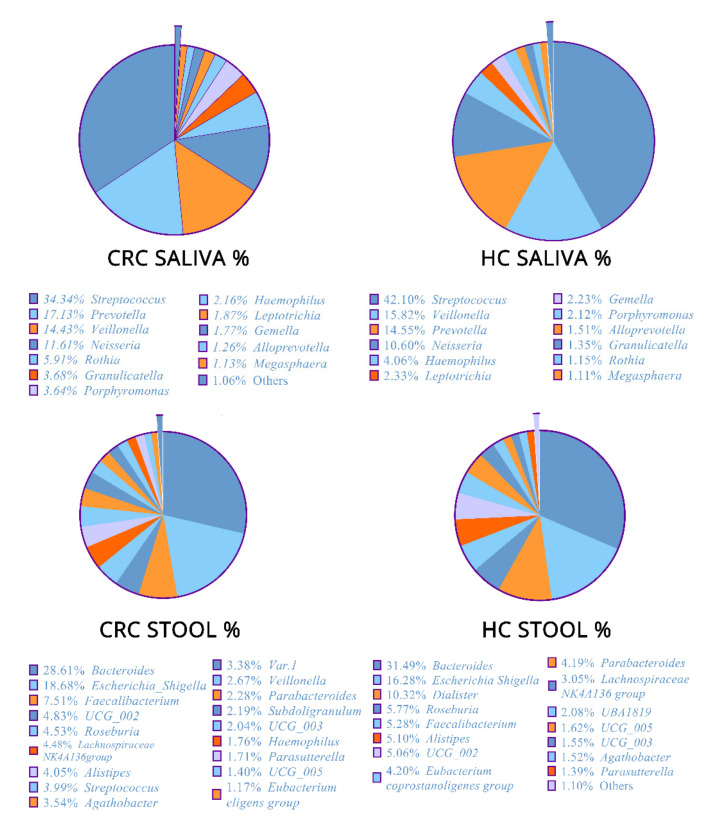
Composition of bacteria and the relative differential abundance of the bacterial genera in fecal and saliva samples of colorectal cancer (CRC) patients and healthy controls (HCs).

**Figure 2 cancers-15-00192-f002:**
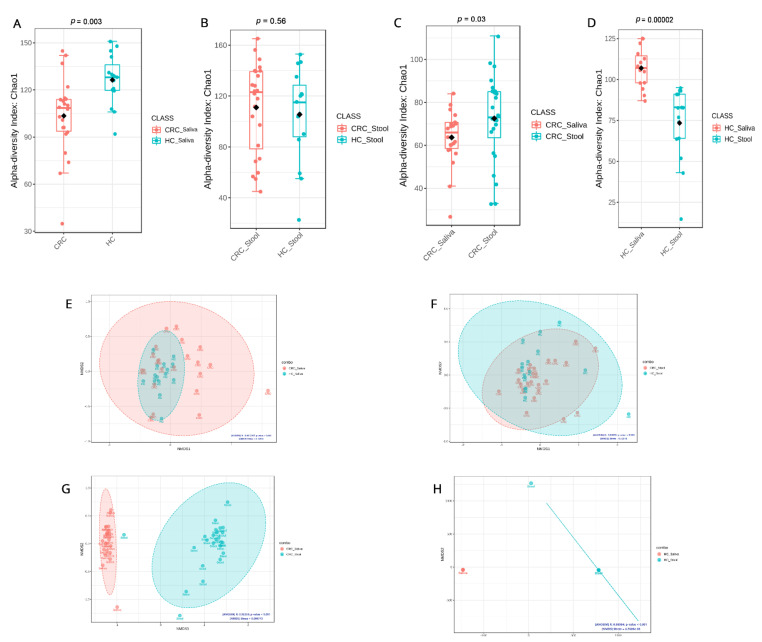
Microbial community diversity across our study groups using the Chao1 approach for α-diversity and NMDS metric for β-diversity: (**A**) α-diversity of saliva samples from CRC patients and healthy controls; (**B**) α-diversity of stool samples from CRC patients and healthy controls; (**C**) α-diversity of saliva and stool samples from CRC patients; (**D**) α-diversity using CHAO1 of saliva and stool samples from healthy controls; (**E**) β-diversity of saliva samples from CRC patients and healthy controls; (**F**) β-diversity of stool samples from CRC patients and healthy controls; (**G**) β-diversity of saliva and stool samples from CRC patients; (**H**) β-diversity of saliva and stool samples from healthy controls.

**Figure 3 cancers-15-00192-f003:**
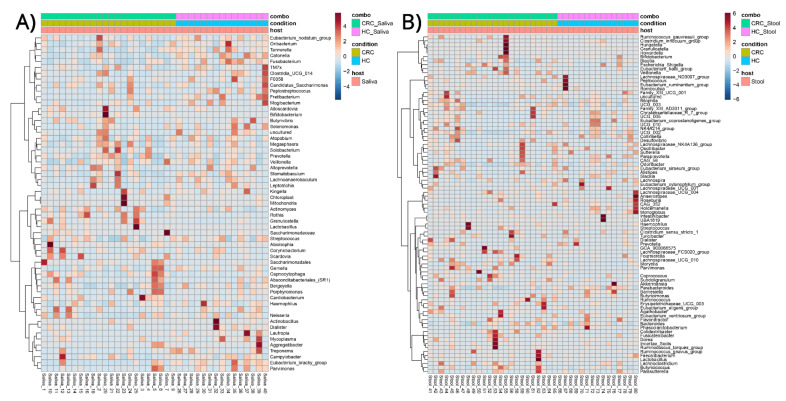
Clustered heatmap of bacterial genera–sample group associations in (**A**) saliva samples of colorectal cancer (CRC) patients and healthy controls (HCs) and (**B**) stool samples of colorectal cancer (CRC) patients and healthy controls (HCs).

**Figure 4 cancers-15-00192-f004:**
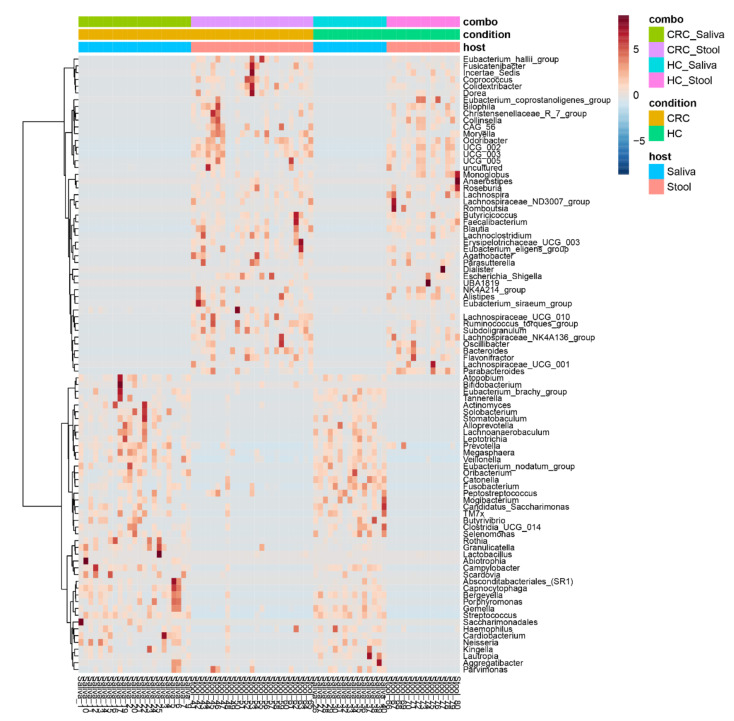
Clustered heatmap of bacterial genera–sample group associations regardless of explanatory variable. The heatmap shows a clear distinction between saliva and stool samples while the disease creates different association patterns versus healthy controls.

**Figure 5 cancers-15-00192-f005:**
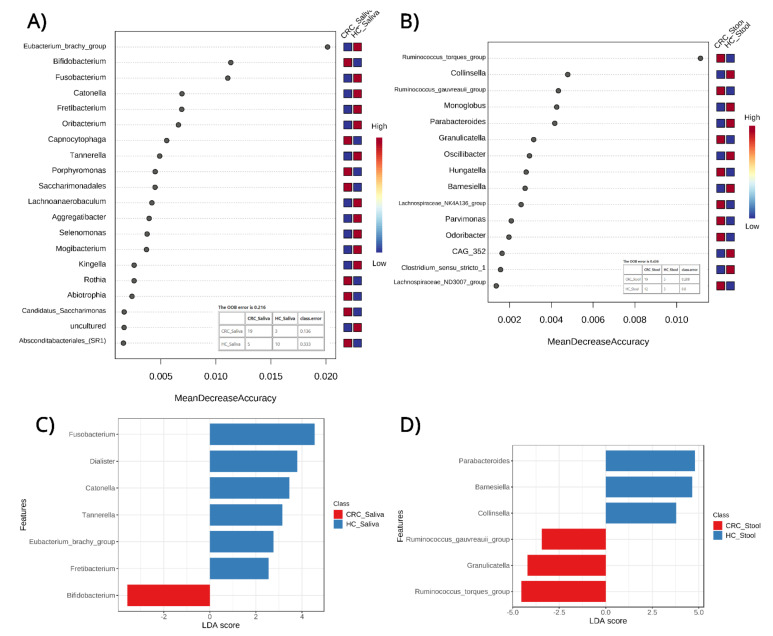
Top features (bacterial genera) in Random Forest (RF) models using (**A**) saliva samples of CRC (Colorectal cancer) patients and healthy controls (HC); (**B**) stool samples of CRC patients and healthy controls to predict sample classification into the patient and control groups. Figures also include Out-Of-Bag error and classification matrices for each model. In addition, Linear Discriminant Analysis (LDA) (**C**,**D**) between the same sample groups reveals bacterial genera which can serve as biomarkers for possible sample classification in CRC. Both approaches highlight, through different statistical methodologies, specific bacterial genera. ●.

## Data Availability

Most data generated and analyzed during this study is included in this published article. Additional dataset used and/or analyzed during the current study are available from the corresponding author on reasonable request.

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
