# Peer review of "Oral Microbiota as Novel Biomarkers for Colorectal Cancer Screening"

_cancers, 2022, doi:10.3390/cancers15010192_

Round 1

Reviewer 1 Report

Dear Authors

I appreciate your efforts in this study.

This study attempted to evaluate whether the oral and fecal micro biomes of an individual are suitable for CRC screening.

The subject is interesting and may lead to clinical implications in the future, the design and description of the study and discussion of the results was done well. However, there are minor shortcomings that I suggest this manuscript is revised  .

I added my comments to the text of manuscript.

With Regards

Reviewer 2 Report

Thank you for the opportunity to review this interesting manuscript. Many studies have suggested that the gut microbiome may be an important factor in the development of colorectal cancer. The using novel microbiome biomarkers of colorectal cancer in concert with known clinical risk factors could improve the ability to identify candidates for colonoscopy. In this work the aim of the authors was to utilize metagenomics analyses in order to compare the oral and fecal microbiome composition and diversity between CRC patients and healthy control (HC) in the Iranian population.

I have some suggestions to improve this manuscript. In the abstract it is necessaty to specify the aim of the study. In the introduction, I find  inadequate the sentence "Colorectal cancer (CRC) is considered one of the most important malignancies". A scientific connotation must be provided. I can be useful an epidemiological focus on CRC. One of the most commonly used noninvasive screening procedures is the FIT. I think is necessary to explain the opportunity to research new strategy of CRC screening. 

The methodology is well structured. 

In the discussion, it would be useful to better discuss why it is necessary to find mew screening strategies. Are the coverage indicators sufficient worldwide? What are the main barrieres of this screening? The current test is not well accepted by the population?

Round 2

Reviewer 2 Report

This version is ok for me.

Author Response

Thank you for your positive comments